# Examining Gender Bias of Convolutional Neural Networks via Facial Recognition

Tony Gwyn * 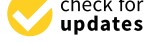 and Kaushik Roy

Department of Computer Science, North Carolina A&T State University, Greensboro, NC 27411, USA
* Correspondence: tgwyn@aggies.ncat.edu

**Abstract:** Image recognition technology systems have existed in the realm of computer security since nearly the inception of electronics, and have seen vast improvements in recent years. Currently implemented facial detection systems regularly achieve accuracy rates close to 100 percent. This includes even challenging environments, such as with low light or skewed images. Despite this near perfect performance, the problem of gender bias with respect to accuracy is still inherent in many current facial recognition algorithms. This bias needs to be addressed to make facial recognition a more complete and useful system. In particular, current image recognition system tend to have poor accuracy concerning underrepresented groups, including minorities and female individuals. The goal of this research is to increase the awareness of this bias issue, as well as to create a new model for image recognition that is gender independent. To achieve this goal, a variety of Convolutional Neural Networks (CNNs) will be tested for accuracy as it pertains to gender bias. In the future, the most accurate CNNs will then be implemented into a new network with the goal of creating a program which is better able to distinguish individuals with a high accuracy, but without gender bias. At present, our research has identified two specific CNNs, VGG-16 and ResNet50, which we believe will be ideal for the creation of this new CNN algorithm.

**Keywords:** convolutional neural networks; gender; gender bias; authentication; presentation attack; presentation attack detection; biometrics; face biometrics; facial detection; facial recognition; classification methods

## 1. Introduction

For decades, image recognition technology has made leaps and bounds in the ability to identify individuals in increasingly challenging conditions. We have even seen this technology move into the medical realm, where it is being used for the diagnosis of various different symptoms, from gastrointestinal lesions [1], to diabetic retinopathy [2] with very effective success. Facial biometrics show great promise with regard to authentication systems. This is due to the fact that the human face has specific facial landmarks that allow for systems to accurately discern subjects [3,4].

While it would be nice to believe that such security methods are unnecessary or overkill, the reality is that while computer authentication methods have become more advanced over the years, the methods with which computers are attacked have also become more sophisticated. In addition, if an intruder is able to obtain an individual's username and password through nefarious means, then they will be able to gain access to that user's information, regardless of the strength of that user's credentials. The reality is that the simple username/password standard is not a reliable level of security to dissuade attacks on computer systems [5,6]. Moving into the future, we need to be able to establish systems that make use of different and more effective authentication methods which can use biometrics and/or facial recognition techniques [7–9].

This brave new world of facial authentication has shown much promise and has garnered more than its fair share of praise in recent years, However, even the most state-of-the-art facial recognition techniques fall short, specifically where darker complexion

individuals and female presenting users are concerned [10]. One study in particular has shown that false positive rates for Western and Eastern African individuals were much higher than that for their Eastern European counterparts [11]. No one can deny the reach of facial recognition technology, and its capacity to do good things in the world; however, we need to make certain that we do everything in our power to mitigate any bias in these systems, to make sure that this technology benefits all people fairly.

The research proposed in this paper will make use of 8 different advanced Convolutional Neural Networks (CNNs), along with 3 different image datasets to measure the accuracy of those CNNs in recognizing faces from differing genders. This paper will also determine which CNN is the most fairly balanced when it comes to any inherent bias. More information on the CNN models and the datasets utilized in this study can be found in Sections 2.1 and 2.2, respectively.

Additionally, research into gender bias in facial recognition lacks behind where other areas of research are concerned. The issue of gender bias in facial recognition systems is not that well documented or investigated thus far. As such, this research will ideally lead to increased awareness of the issue of gender bias inherent in some facial recognition technology.

The accuracy level of the CNNs that are involved in these experiments are well known. However, the overarching purpose of this paper is to examine the potential for those CNNs to have inherent biases, and how we can best deal with those. As such, obtaining the highest possible accuracy in each of our tests is of secondary concern; investigating and mitigating any gender bias is our main objective.

Finally, as a result of this research, we hope to propose a new CNN model, which will combine some of the best performing CNNs tested in this paper. The purpose of this new CNN model will be to successfully mitigate inherent gender bias, while maintaining an impressive level of accuracy and performance.

*Literature Review*

This experimental study on facial recognition leans heavily on research and information that has come before. There are decades of documents concerning image technology available on various platforms that investigate all different types of algorithms and models for the purpose of facial identification and recognition. With so much information available, we decided to focus our research on Local Binary Patterns (LBP) and other algorithms associated with that method [12–14]. However, we came to the conclusion that while LBP was indeed impressive, its efficacy paled in comparison to several newer methods of facial recognition available to us in the literature [15].

Eventually, we moved on to investigating various Convolutional Neural Network (CNN) models. Different CNN architectures have been around for years, but it is only recently that many have seen inroads into the realm of facial identification and recognition, with results that speak for themselves. The amount of CNNs available to us for research was quite overwhelming. The accuracy levels and efficiency of several CNN models are impressive, and have steadily increased in recent years. Indeed, there are a multitude of CNNs that exist in the literature which have seen great success in the realm of facial identification and recognition [16–19].

Several other papers helped shape this manuscript. Schoneveld's work on facial expression recognition [20] was particularly impressive; his work in creating a new deep learning-based approach was extremely useful with respect to our own research into the topic. The specific feature extraction scheme used by Schoneveld's group will be considered for our future work with deep networks.

Hazourli's team used a novel approach for facial image classification that involved deep features interspersed with convolutional neural networks [21]. While this work was primarily most effective with smaller datasets for training, it could in theory be utilized even with deeper neural networks and larger datasets, such as those that were utilized in our experiments.

Emotion recognition is an emerging field that leverages human reaction, along with verbal and non-verbal communication [22]. This complex behavior, when studied with other physical features, can lead to higher accuracy, as well as the potential for a marked increase in robustness of the system. These emotion recognition advances could be extremely useful for future work in this field, specifically with datasets that are tuned to make extensive use of the technology.

As our research progressed, we decided to select a total of 8 different CNN models which are completely foreign to each other. In other words, we were trying to determine which models are best able to handle gender bias. We decided that we needed to choose a variety of CNN models from different families, rather than 8 models that used virtually the same architecture. It was our hope that this variety in our models would help us to create a new hybrid model consisting of the best parts of several CNNs, which would retain the accuracy and performance that CNNs are known for, while simultaneously being more resistant to different genders of tested individuals.

## 2. Materials and Methods

The study contained in this paper was successfully conducted by use of the Python programming language; specifically, Python 3.7 using the 32-bit architecture. This version of Python allowed for the creation of an algorithm that was specifically customized and tailored for image recognition. This Python program utilizes a total of 8 different popular CNN models. These CNN models are all optimized for maximum accuracy and performance, especially as it pertains to gender bias. Again, the primary goal of this work was to find specific CNN models that are most resistant to gender bias, for the purpose of the creation of a new 'novel' CNN model.

In addition to the CNN models utilized, a total of 3 distinct image datasets were also integrated and tested. These image datasets were both implemented and subsequently examined and tested separately, in an effort to make sure that our results were not influenced or biased. The results from each dataset need to firmly stand on their own, without any influence from the other datasets. As such, the accuracy and performance of each CNN may vary slightly between datasets, but the end result will prove to be more robust due to these precautions.

### 2.1. CNN Variants Used

As mentioned, the amount of CNN variants available in the literature for use in image recognition or classification is quite expansive [16,23]; however, the amount of testing and research conducted has led us to conclude that the CNN models included in this paper are among the best considering both accuracy and the mitigation of gender bias [24,25]. These CNN variants include a total of 8 distinct Deep Net models, which include AlexNet, Xception, two versions of VGG (VGG-16 and VGG-19), two versions of ResNet (ResNet50 and ResNet101), and two versions of Inception (Inception v2 and v3). A brief comparison of each of these different CNN variants can be found in Table 1, including the architecture version used, as well as the total trainable parameters of each CNN, and the feature that makes that particular method stand out from the crowd. More details on each of these CNNs can be found in Section 3.1.1 through Section 3.1.5.

**Table 1.** Comparison of different CNN Models used in our study.

| CNN Model | ArchVersion | Total Layers | Conv Layers | Trainable Params | Unique Features |
|---|---|---|---|---|---|
| AlexNet | Version 1 | 8 | 5 | 62,378,344 | ReLU activation |
| Xception | Version 1 | 71 | 36 | 22,855,952 | Depth-Seperable Convs |
| Inception v2 | Version 2 | 48 | 22 | 55,813,192 | Wider-Parellel Kernels |
| Inception v3 | Version 3 | 48 | 22 | 23,817,352 | Wider-Parellel Kernels |
| ResNet50 | Version 1 | 50 | 48 | 25,583,592 | Simpler Mapping |
| ResNet101 | Version 2 | 101 | 99 | 44,601,832 | Simpler Mapping |

**Table 1.** *Cont.*

| CNN Model | ArchVersion | Total Layers | Conv Layers | Trainable Params | Unique Features |
|-----------|-------------|--------------|-------------|------------------|-----------------|
| VGG-16 | Version 1 | 16 | 13 | 138,357,544 | Fixed-size Kernels |
| VGG-19 | Version 2 | 19 | 16 | 143,667,240 | Fixed-size Kernels |

*2.2. Datasets Used*

With the CNN models established, we needed to decide on which image datasets we should include for our study. We settled upon 3 different publicly available video datasets, each with a very impressive track record that we could utilize very effectively to test our CNNs accuracy and bias mitigation. For clarity, as we are investigating image recognition with our program, we take still images from each of these video datasets and use them to examine the accuracy and performance of each of our CNN models.

These video datasets include the Wide-Multi Channel Presentation Attack (WMCA) Dataset [26], the CASIA Face Anti-Spoofing Dataset (CASIA-FASD) [27], and the Spoofing in the Wild (SiW) Database [28]. A brief comparison of each of these datasets can be found in Table 2, including the respective dates the videos were collected, as well as the number of videos and subjects included in each dataset. Further details about each dataset can be found in Section 3.2.1 through Section 3.2.3.

**Table 2.** Comparison of different datasets used in our study.

| Dataset | Year Collected | Total Subjects | Total Videos | Images Extracted | Total Size |
|---------|----------------|----------------|--------------|------------------|------------|
| WMCA | 2019 | 72 | 1941 | 3000+ | 1.1+ GB |
| CASIA-FASD | 2015 | 50 | 600 | 2500+ | 950+ MB |
| SiW | 2018 | 165 | 4478 | 4200+ | 1.6+ GB |

According to the previous table, the number of images extracted from each dataset varied quite significantly. This is due to the fact that the number of videos in each dataset was different as well. In addition, the total amount of subjects was also varied, with the Spoofing in the Wild dataset having over 3 times the number of subjects as the CASIA Face Anti-Spoofing Dataset. Measures were taken to make sure the datasets were as equitable as possible; however, to safeguard against any bias, each of the datasets were examined and tested separately from each other to obtain accurate measurements for each CNN model in our study.

*2.3. Python Environment*

As mentioned, the Python programming language was the ideal candidate for the testing and implementation of our work. The PyCharm IDE worked quite well with both our datasets and our chosen CNN models; in addition, we also imported several different Python packages to help with our image classification. While we did tweak and enhance the various CNN models that were utilized for testing purposes, the initial CNN models were obtained via the Keras open-source software library. A visualization of the implementation of the base CNN models can be seen in Figure 1. It was decided that we should utilize Keras due to its built-in functionality for interfacing with various neural network architecture [29].

```python
from keras.applications.vgg16 import VGG16
from keras.applications.vgg19 import VGG19
from keras.applications.resnet import ResNet50
from keras.applications.resnet import ResNet101
```

**Figure 1.** CNN models initialized by Keras open-source library.

*2.4. Program Structure*

We utilized various Python packages that are typical for image recognition and application; these included Pandas, Tensorflow, as well as ImUtils. Once those packages were successfully imported, we moved on to the next step, which included introducing the base versions of the CNN models that we had selected previously, which can be seen in Figure 1. The base models of these CNN models are powerful and highly accurate in their own right, and one of their main purposes is dedicated to the detection and classification of images. However, it was decided that we should attempt to enhance these models to be both highly accurate, as well as highly resistant to gender bias, in an attempt to obtain the best results.

After the CNN models were introduced, we began the process of introducing our datasets into the program. MatPlotLib is a Python package that is synonymous with graphing, and so it was utilized for that specific purpose in our study. After this step, we next imported Python functions that would be vital for the training and testing of our model, in addition to functions that would be utilized for 'scoring' our CNN models based on accuracy metrics and overall model performance.

Once the datasets and various Python functions were added to the program, the training of the models was set to begin. Each dataset was tested separately to ensure no bias could transfer between datasets. After the training and testing phase was concluded for a single dataset, the results for the CNNs were examined based on their accuracy and performance. All accuracy data was documented and removed from the system in order to introduce and test the next dataset. This continued until each CNN model had been tested against each dataset, and all of the datasets were exhausted.

## 3. Methodology

During the course of our testing, the CNN models had their accuracy and performance documented using various metrics. Additionally, their ability to handle differing genders was also recorded, as that ability is especially desirable for the purpose of our study. Any CNN models that showed exemplary performance in our metrics will be put forth as potential candidates for a hybrid CNN model. Keeping a high level of accuracy and performance, along with an increased amount of gender bias mitigation, is the ideal end goal of our current research.

*3.1. Convolutional Neural Networks Detailed*

Convolutional Neural Networks (CNNs) are a specific type of Neural Network which has seen great advances and admirable performance in the area of Computer Vision and Image Recognition [30]. An example of a simple CNN model with 4 different potential labels can be seen Figure 2; this example shows the predicted output of the sample image as being that of the cat label, with 94 percent confidence.

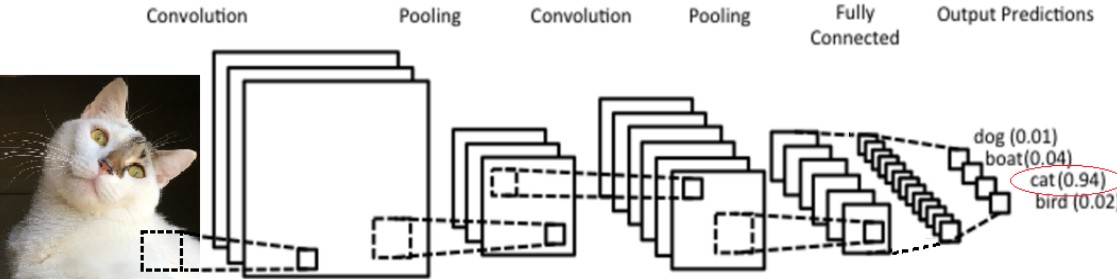

**Figure 2.** Basic Convolutional Neural Network.

CNNs are also being applied to various other areas, including Natural Language Processing (NLP), Speech Recognition, and Video Processing [31,32]. Several papers have been made on these specific areas concerning CNNs; however, this paper will focus on the application of CNNs for Image Classification. We also specifically want to examine their ability to solve the pervasive issue of gender bias that is inherent in some classification models.

### 3.1.1. AlexNet

Convolutional Neural networks have recently become the standard for image and object recognition; however, this does not mean that they are infallible. One of their shortcomings is that some CNNs have difficulty in processing higher resolution images, which is an obvious problem for a model that is looking for specific features in an image to classify it. AlexNet, named for its creator Alex Krizhevsky, was created in 2012 in an attempt to mitigate this issue by optimizing its model using Graphics Processing Units (GPUs). Later that year, AlexNet was already winning praise by competing and placing quite well in the ImageNet Large Scale Visual Recognition Challenge (ILSVRC).

Along with fellow researchers Ilya Suskever and Geoffrey Hinton, Dr. Krizhevsky used Rectified Linear Units (ReLU) in addition to multiple GPUs and a process known as overlapping pooling to better handle larger resolution images. Due to this novel process, AlexNet is able to attain much better training times on datasets, as well as a greatly reduced amount of processing errors, even when larger datasets are utilized [33]. A recreation of the original AlexNet architecture can be viewed in Figure 3, along with the overlapping pooling method.

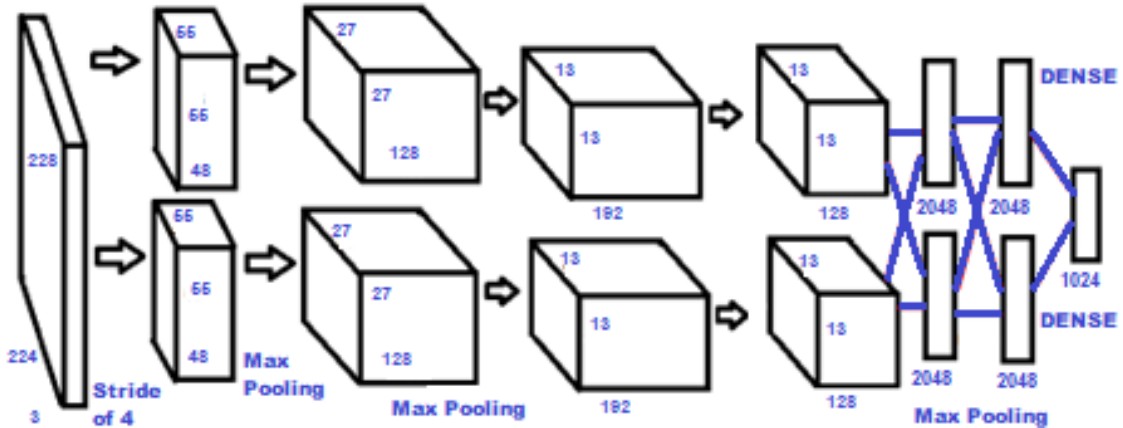

**Figure 3.** AlexNet Neural Network, based on the original architecture.

### 3.1.2. Inception v2/Inception v3

While Convolutional Neural Networks are still seen as a 'newer' technology in the field of image classification, it still has a varied history. Before the creation of the Inception architecture, stacking convolution layers was the order of the day. It was believed that this was the best and easiest way to attain high performance and increased accuracy; however, the Inception model tries a different approach. Instead of taking the quick and easy route, Inception was engineered, leading to a quick evolution of its models, including several different versions that are still widely used today [34].

The hope of Inception was to solve issues with some images that would vary in size, depending on the salient parts of those images. The inherent variation caused problems with models that were not robust, leading to issues with both model performance and overall image recognition accuracy. CNNs needed to use the correct kernel size for convolutions compared to the images that they were working with. In example, a smaller kernel size is useful for dealing with image information that is local (say, a part of an image), whereas a larger kernel size was preferable when dealing with an image that was global (where the image information consisted of the entire image).

Inception successfully solved this issue by having different filters with multiple sizes. Instead of utilizing deeper networks, the Inception architecture opted to go wider with its convolutions. While there are several different Inception models widely in use today (with even more still being developed and investigated), we decided to go with Inception version 2 and Inception version 3 with our research paper, due both to their performance, and to their very recent use in several research papers [34–36].

Inception version 2 improved upon the initial Inception architecture by making the wider convolutions more efficient, as well as attempting to alleviate potential bottlenecks which could lead to a loss of information. Inception version 3 introduced novel 7-by-7 width parallel convolutions, and also tweaked some of the auxiliary classifiers. Due in part to these changes, while both versions of Inception share various similarities, they are able to have image classification accuracy and performance that is sometimes widely varied [35]. An example of this architecture can be seen in Figure 4.

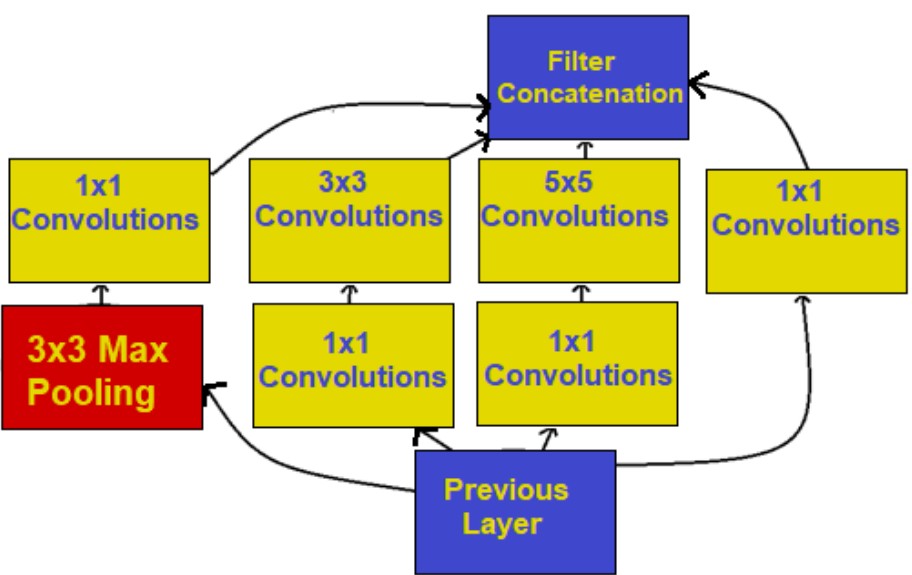

**Figure 4.** Inception version 2 architecture with dimension reductions, based on the original architecture.

### 3.1.3. Xception

This unique Convolutional Neural Network is partially adapted from the Inception Model, several versions of which are included in our research. While Inception typically uses modules that go wider instead of deeper, Xception uses convolutions that have depthwise separability. This means that although Xception has nearly the same overall parameters of Inception v3 due to their very similar parameters and architecture, that very different results can often be obtained by the two models. A recreation of the original Xception model cam be seen in Figure 5 [37].

Xception is often considered the 'black sheep' of the Inception family, due to the fact that it tends to take some of the ideas established by the Inception architecture to their logical and perhaps extreme conclusion [38]. Inception versions tend to use 1-by-1 convolutions, followed by either 3-by-3 or 5-by-5 convolutions that handle certain special correlations. By contrast, Xception instead performs 1-by-1 convolutions for every channel first, followed by a 3-by-3 calculation which is added to each of the outputs. This is the defining feature of Xception, which is what creates the specific depthwise separable convolutions that it is known for, making it unique among most other CNN models.

### 3.1.4. VGG-16/VGG-19

The initial VGG architecture was proposed back in 2014, by Karen Simonyan and Andrew Zisserman, both of the world-famous Oxford Robotics Institute. Short for Visual Geometry Group, both VGG-16 and VGG-19 are extremely high pedigree CNN models that specialize in both image classification, as well as object localization [24]. Much like its compatriot AlexNet, VGG-16 competed and performed quite well in the 2014 ILSVRC, where it placed second in the image classification challenge, and attained a stunning first place in the object localization challenge.

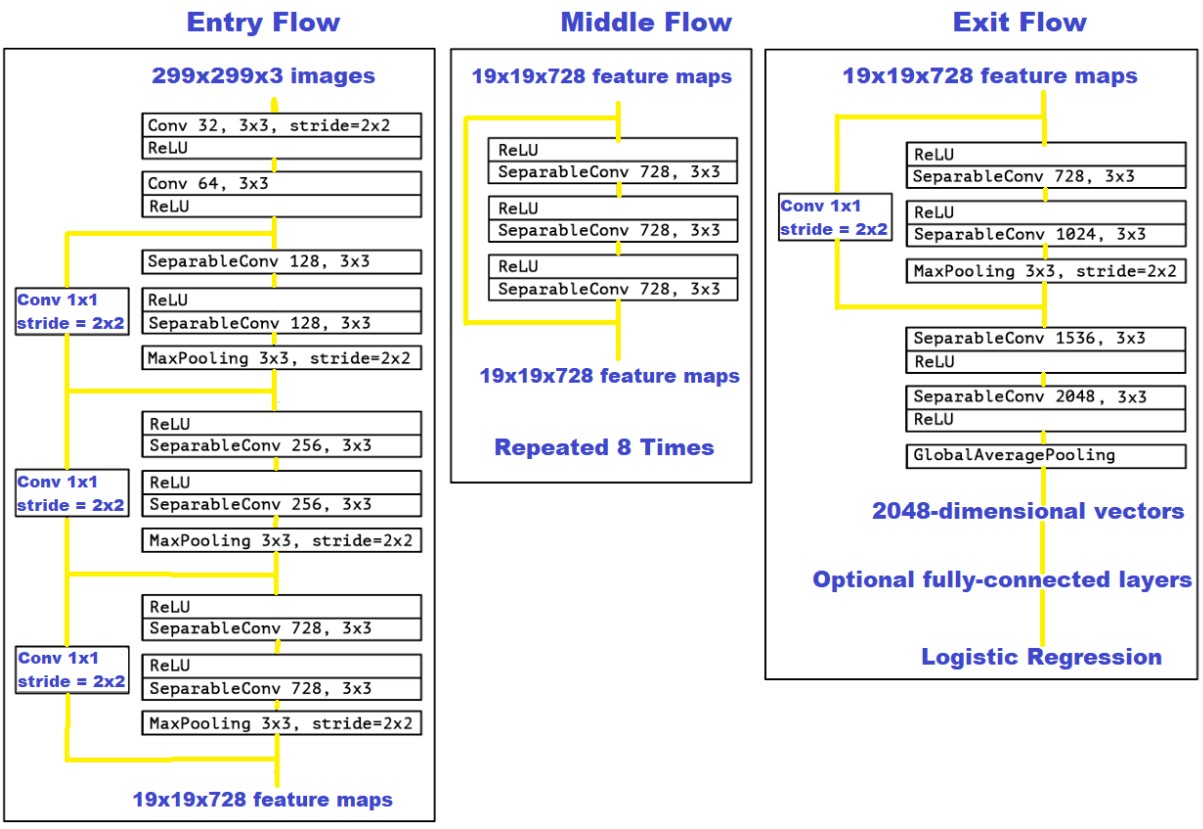

**Figure 5.** Xception with entry, middle, and exit flow defined, based on the original architecture.

VGG is perhaps the most well-known of all of the different CNN model groups, due to the impressive results it has been able to obtain nearly since its inception into the field. VGG uses 3-by-3 convolutions, and obtains quite impressive accuracy, even with an architecture that is smaller than most similar CNN models. While fairly similar, the main difference between VGG-16 and VGG-19 consists of the total number of depth layers in each architecture; of which the majority of those are convolution layers, which should not come as a surprise. A recreation of the initial VGG architecture, which includes both the VGG-16 and VGG-19 model, can be viewed in Table 3 [39].

**Table 3.** VGG-16 and VGG-19 models, based on the original architectures. Added layers have been denoted in bold.

| ConvNet Configuration | | | | | |
|---|---|---|---|---|---|
| A | A-LRN | B | C | D | E |
| 11 weight layers | 11 weight layers | 13 weight layers | 16 weight layers | 16 weight layers | 19 weight layers |
| input (224 × 224 RGB Image) | | | | | |
| conv3-64 | conv3-64 **LRN** | conv3-64 **conv3-64** | conv3-64 conv3-64 | conv3-64 conv3-64 | conv3-64 conv3-64 |
| maxpool | | | | | |
| conv3-128 | conv3-128 | conv3-128 **conv3-128** | conv3-128 conv3-128 | conv3-128 conv3-128 | conv3-128 conv3-128 |
| maxpool | | | | | |
| conv3-256 | conv3-256 | conv3-256 | conv3-256 | conv3-256 | conv3-256 |

**Table 3.** *Cont.*

| ConvNet Configuration | | | | | |
|---|---|---|---|---|---|
| conv3-256 | conv3-256 | conv3-256 | conv3-256 **conv1-256** | conv3-256 **conv3-256** | conv3-256 conv3-256 **conv3-256** |
| maxpool | | | | | |
| conv3-512 conv3-512 | conv3-512 conv3-512 | conv3-512 conv3-512 | conv3-512 conv3-512 **conv1-512** | conv3-512 conv3-512 **conv3-512** | conv3-512 conv3-512 conv3-512 **conv3-512** |
| maxpool | | | | | |
| conv3-512 conv3-512 | conv3-512 conv3-512 | conv3-512 conv3-512 | conv3-512 conv3-512 **conv1-512** | conv3-512 conv3-512 **conv3-512** | conv3-512 conv3-512 conv3-512 **conv3-512** |
| maxpool | | | | | |
| FC-4096 | | | | | |
| FC-4096 | | | | | |
| FC-1000 | | | | | |
| soft-max | | | | | |

During our research into VGG, it was clear that we needed to include at least several different models for our research due to its impressive accuracy and ability to handle challenging image conditions. As such, we settled on the two with the best track record where accuracy and performance are concerned; namely, VGG-16 and VGG-19. Although they are both best-in-class when it comes to image classification, there are several difficulties with the VGG architecture that can make it a challenge to successfully utilize.

VGG at its core is quite robust; while this can be a boon when training and testing a myriad of image databases, this can also make VGG quite difficult to train without several powerful GPUs. In addition to this, when VGG was first being investigated, some of the weighted that VGG used for calculations caused the models to use an inordinate amount of bandwidth and disk space, although those issues have mostly been mitigated due to recent advances in storage capabilities of computer systems [40].

### 3.1.5. ResNet50/ResNet101

In recent years, the task of classifying images has become more and more complex, and CNN models are expected to become more accurate, with better performance and the ability to be more robust. The answer at first was to simply create models that were able to utilize neural networks that moved deeper and further to meet these challenges. However, this answer posed its own challenges; these deeper neural networks were harder to train. Not only that, but due to that difficulty, there was an overall degradation in model accuracy, which caused some architectures to have to return to square one.

Introduced in 2015, the ResNet architecture, established by Kaiming He, Xiangyu Zhang, Shaoqing Ren, and Jian Sun, was established to help mitigate this new found problem. Residual Network (ResNet) is a CNN which is able to maintain and increase performance and accuracy [41], even while stacking these additional convolutional and depth layers. More and more complex features are able to be identified and examined as the deeper layers are added, which correlates to better system performance, as well as the obvious boon of increased accuracy and system robustness. However, a delicate balance does need to be struck between the number of layers in a ResNet model, as too many layers can cause an overall decrease in accuracy, along with an increase in the overall

error percentage. Sometimes, a ResNet will fewer layers that is implemented successfully will outperform a similar ResNet, even with more layers added [42].

Specifically in our research, ResNet50 and ResNet101 showed themselves to be ideal candidates for inclusion in our study, as it seemed that they both were able to strike the delicate balance between too many depth layers and a reduction in performance and accuracy. ResNet50 is considered the de facto version of the ResNet architecture, while ResNet101 itself has gone up against several heavy hitters in the realm of image classification such as the Inception architecture and VGG-16, acquitting itself quite nicely in the process [43]. An example of the initial ResNet architecture, containing both architectures used in our study, can be found in Table 4.

**Table 4.** A variety of ResNet models, based on their original architectures.

| Layer Name | Output Size | 18-Layer | 34-Layer | 50-Layer | 101-Layer | 152-Layer |
|---|---|---|---|---|---|---|
| conv1 | $112 \times 112$ | \multicolumn{5}{c}{$7 \times 7$, 64, stride 2} |
| | | \multicolumn{5}{c}{$3 \times 3$ max pool, stride 2} |
| conv2.x | $56 \times 56$ | $[3 \times 3, 64]$<br>$[3 \times 3, 64] \times 2$ | $[3 \times 3, 64]$<br>$[3 \times 3, 64] \times 3$ | $[1 \times 1, 64]$<br>$[3 \times 3, 64] \times 3$<br>$[1 \times 1, 256]$ | $[1 \times 1, 64]$<br>$[3 \times 3, 64] \times 3$<br>$[1 \times 1, 256]$ | $[1 \times 1, 64]$<br>$[3 \times 3, 64] \times 3$<br>$[1 \times 1, 256]$ |
| conv3.x | $28 \times 28$ | $[3 \times 3, 128]$<br>$[3 \times 3, 128] \times 2$ | $[3 \times 3, 128]$<br>$[3 \times 3, 128] \times 4$ | $[1 \times 1, 128]$<br>$[3 \times 3, 128] \times 4$<br>$[1 \times 1, 512]$ | $[1 \times 1, 128]$<br>$[3 \times 3, 128] \times 4$<br>$[1 \times 1, 512]$ | $[1 \times 1, 128]$<br>$[3 \times 3, 128] \times 8$<br>$[1 \times 1, 512]$ |
| conv4.x | $14 \times 14$ | $[3 \times 3, 256]$<br>$[3 \times 3, 256] \times 2$ | $[3 \times 3, 256]$<br>$[3 \times 3, 256] \times 6$ | $[1 \times 1, 256]$<br>$[3 \times 3, 256] \times 6$<br>$[1 \times 1, 1024]$ | $[1 \times 1, 256]$<br>$[3 \times 3, 256] \times 23$<br>$[1 \times 1, 1024]$ | $[1 \times 1, 256]$<br>$[3 \times 3, 256] \times 36$<br>$[1 \times 1, 1024]$ |
| conv5.x | $7 \times 7$ | $[3 \times 3, 512]$<br>$[3 \times 3, 512] \times 2$ | $[3 \times 3, 512]$<br>$[3 \times 3, 512] \times 3$ | $[1 \times 1, 512]$<br>$[3 \times 3, 512] \times 3$<br>$[1 \times 1, 2048]$ | $[1 \times 1, 512]$<br>$[3 \times 3, 512] \times 3$<br>$[1 \times 1, 2048]$ | $[1 \times 1, 512]$<br>$[3 \times 3, 512] \times 3$<br>$[1 \times 1, 2048]$ |
| | $1 \times 1$ | \multicolumn{5}{c}{average pool, 1000-d fc, softmax} |
| FLOPs | | $1.8 \times 10^9$ | $3.6 \times 10^9$ | $3.8 \times 10^9$ | $7.6 \times 10^9$ | $11.3 \times 10^9$ |

### 3.2. Datasets Detailed

Image datasets are a key part of any investigation into the efficacy of image classification architecture. Video datasets are often used in the place of image datasets, as a number of images can be extracted from each video; in addition, images of an individual taken from a video will be similar, but not exactly the same for that subject, which will ideally lead to increased robustness and accuracy for our CNN models. As mentioned previously, in this study, we make use of 3 distinct video datasets; details about each of those can be found in the next few sections of this paper.

For clarification, each of the image datasets mentioned in the following sections are governed by their own specific license or agreement. The individuals portrayed in the following figures agreed to have their likenesses used for those datasets. Specifically, the WMCA Dataset is governed by an End-User License Agreement (EULA) restricted to non-commercial research, while the CASIA-FASD Dataset uses an Open-Source Initiative (OSI) approved Berkeley Source Distribution (BSD) License. Finally, the SiW Database is available for use via a license obtained via Michigan State University for research purposes.

### 3.2.1. Wide Multi Channel Presentation Attack (WMCA)

The WMCA Dataset was first created in 2019, by Anjith George and their team. As is typical for datasets that deal with presentation attacks, the WMCA dataset has data from both genuine, as well as falsified individuals [26]. It is of note that this particular dataset

uses data which is recorded from several different channels, which include thermal, color, depth, infrared, as well as others.

This dataset contains a total of 1941 short video recordings, which are comprised of a total of 72 distinct individuals (both genuine and falsified). In total, we generated over 3000 images from this dataset, for use with training and testing our various CNN models, as well as determining their overall performance, robustness, and ability to mitigate gender bias. Some samples images from the Wide Multi Channel Presentation Attack Dataset can be seen in Figure 6.

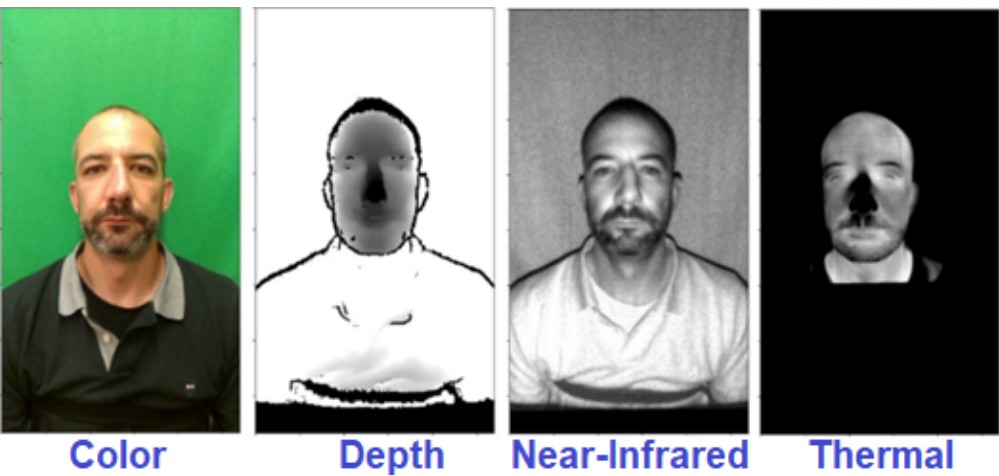

**Figure 6.** Sample images of an individual, recreated from the WMCA Dataset.

### 3.2.2. CASIA Face Anti-Spoofing Dataset (CASIA-FASD)

The CASIA Dataset is an anti-spoofing dataset, that is primarily compromised of facial images from different individuals, both genuine and fake [27]. Established by Zwihei Zhang and their team in 2015, it is one of the more famous facial datasets created for the purpose of anti-spoofing. While the dataset is slightly smaller, comprised of only 50 genuine subjects and 600 total videos, with fake faces being created using high quality images of genuine faces.

In total, for this dataset, we were able to extract 2500 different images from the video collection. As with the other datasets, a number of the individuals in this dataset are genuine, while several others have been generated and are fake. Some sample images from the CASIA-FASD can be seen in Figure 7.

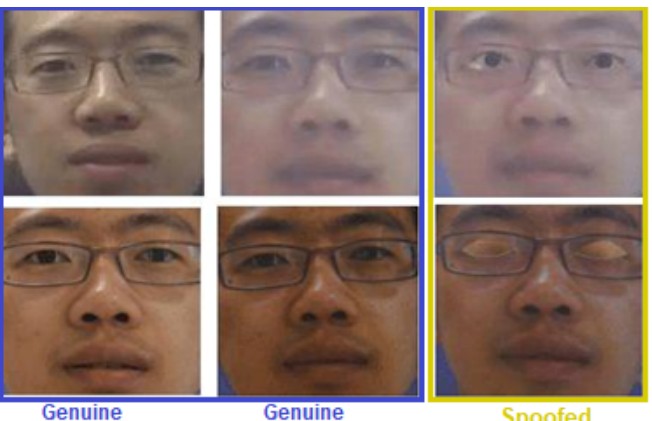

**Figure 7.** Sample images of an individual containing both genuine and generated images, recreated from the CASIA Face Anti-Spoofing Dataset.

### 3.2.3. Spoofing in the Wild (SiW)

Our last dataset is the Spoofing in the Wild (SiW) Database, which was first established in 2018. Created and maintained by Yaojie Liu, Anim Jourabloo and Xiaoming Liu, the SiW Database was initially collected in response to the need to combat various high-quality spoof mediums that had begun to appear during that timeframe [28]. As with the other datasets, the SiW Database contains both live and spoof videos of different subjects.

The SiW Database is the largest of the datasets we have utilized, containing 165 subjects, and a variety of both live and spoof videos of those subjects. The total number of videos in the SiW Database is over 4400; from those videos, we were able to extract over 4200 images for the purpose of testing the abilities of our CNN models. Some examples of images from the SiW Database can be seen in Figure 8; the images in the top row are genuine, while the images in the bottom row have been spoofed.

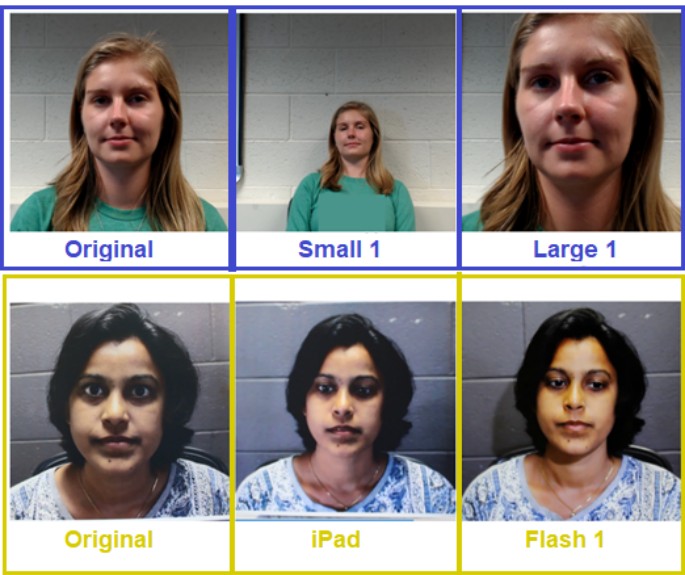

**Figure 8.** Sample images of individuals, recreated from the Spoofing in the Wild (SiW) Database. The individual in the top row is genuine, whereas the individual in the bottom row has been spoofed.

## 4. Discussion

The overarching purpose of this study is to determine which of the tested CNN models are the most effective with respect to gender bias. Those models found to be most resistant to bias would be examined in an effort to create a hybrid CNN model; ideally, maintaining the high level of accuracy and performance of the tested CNNs, whilst also increasing the system's overall ability to mitigate gender bias.

A few items of note: while the original goal of this paper was to examine demographic bias, sufficiently large datasets containing videos or images with diverse candidates were not easily obtained. As such, in this paper we solely investigate the ability for our chosen CNN models to mitigate gender bias instead. In the future, we plan to investigate datasets with labels that include specific demographic terms, and include them in similar studies.

Additionally, while the included datasets contain both genuine and spoofed images, as mentioned, we were primarily interested in examining gender bias in our CNN models. As such, we do not make distinction between genuine and spoofed images, instead using both types of images to test our theory about potential biases that may exist in facial recognition algorithms.

### 4.1. Results

Our results can be found in the following figures. We took the images created from our 3 datasets and created 3 different subsets from each of those. The purpose of creating those subsets was to examine the image recognition accuracy and potential for gender bias for

each CNN model. Each of these subsets was tested 5 separate times per CNN model. After the results were collected, the median accuracy value obtained was added to the following tables. K-fold cross validation is a superior statistical analysis method, and will be utilized for any future testing of these datasets and CNN models.

The three created subsets are Balanced (50% male individuals, 50% female individuals), Male Dominant (80% male individuals, 20% female individuals), and Female Dominant (20% male individuals, and 80% female individuals). For a given CNN model, if the accuracy difference between the male dominant and female dominant subsets is greater than 10%, we consider that there is a high likelihood that the model is affected by gender bias.

For the WMCA Dataset, we were able to generate over 3000 datapoints (extracted images) for use in our study. This number of images was smaller than what was generated by the SiW Database, but larger than the image amount for the CASIA-FASD Dataset. The accuracy level of our models was also split, being mostly higher than CASIA-FASD, but markedly lower than SiW. Finally, several of the CNN models tested showed high likelihood of potential gender bias, with 5 out of 8 models having a gender disparity of greater than 10 percent for the male and female dominant image subsets.

In Table 5, we can clearly see that both VGG models performed well with accuracy; however, the fall in accuracy for VGG19 concerning the female dominant dataset caused that model to be classified as having the potential for gender bias. On the other side of the accuracy scale, both Xception and AlexNet performed the worst of the group; in addition, both of these models did not do well in terms of potential gender bias, with those models having 11 and 12 percent gender disparity, respectively.

**Table 5.** Accuracy of tested CNNs by gender, WMCA Dataset, 80/20 split.

| CNN Model | Balanced | Male Dominant | Female Dominant | Disparity > 10%? |
|-----------|----------|---------------|-----------------|------------------|
| AlexNet | 69% | 73% | 62% | YES |
| Xception | 63% | 67% | 55% | YES |
| Inception v2 | 74% | 78% | 66% | YES |
| Inception v3 | 76% | 79% | 65% | YES |
| ResNet50 | 79% | 80% | 73% | No |
| ResNet101 | 78% | 80% | 72% | No |
| VGG16 | 84% | 85% | 81% | No |
| VGG19 | 82% | 82% | 71% | YES |

For the CASIA-FASD Dataset, we were able to generate over 2500 datapoints (extracted images) for use in our study. This was the smallest number of images generated by any dataset, which could potentially explain the slightly lower accuracy when compared to other datasets in our study. In addition, for the CASIA-FASD Dataset, 5 out of 8 of the CNN models exhibited a propensity for potential gender bias, which tied for the highest percentage of any of our datasets.

Looking at Table 6, the VGG models again scored high marks for accuracy. In this round of testing, both VGG-16 and VGG-19 also showed resistance to potential gender bias, with a gender disparity of 5 and 6 percent, respectively. Unfortunately, both AlexNet and Xception again performed poorly in all of the image subset testing; in addition, they also were shown to likely have problems with diverse datasets, as their gender disparity levels were both at 12 percent.

For the SiW Database, we generated over 4200 datapoints (extracted images), making it the largest dataset by far. We see higher accuracy levels across the board for our CNN models potentially as a result of the higher number of images available for training the CNN models. The SiW Database has a far higher accuracy level than either the WMCA or CASIA-FASD Datasets, and as is visible in Table 6, the disparity between the male dominant and female dominant subsets was less than 10 percent for 5 out of the 8 CNN models.

**Table 6.** Accuracy of tested CNNs by gender, CASIA-FASD Dataset, 80/20 split.

| CNN Model | Balanced | Male Dominant | Female Dominant | Disparity > 10%? |
|---|---|---|---|---|
| AlexNet | 68% | 74% | 62% | YES |
| Xception | 62% | 66% | 54% | YES |
| Inception v2 | 76% | 77% | 65% | YES |
| Inception v3 | 78% | 81% | 70% | YES |
| ResNet50 | 80% | 82% | 75% | No |
| ResNet101 | 77% | 79% | 68% | YES |
| VGG16 | 83% | 83% | 78% | No |
| VGG19 | 84% | 85% | 79% | No |

As can be seen in Table 7, the VGG-16 and VGG-19 models were the clear winners in this dataset study, attaining over 90 percent accuracy for the male dominant image subset. They were also able to achieve a respectable 84 and 85 percent accuracy for the female dominant subset, meaning that the disparity for our gender imbalanced subsets was 7 and 5 percent for each of those CNN models, respectively. Although they had reduced accuracy when compared to the VGG models, both ResNet models as well as Inception version 2 were also able to keep the disparity for the gender imbalanced subsets below 10%, meaning that they handled potential gender bias reasonably well.

**Table 7.** Accuracy of tested CNNs by gender, SiW Database, 80/20 split.

| CNN Model | Balanced | Male Dominant | Female Dominant | Disparity > 10%? |
|---|---|---|---|---|
| AlexNet | 76% | 80% | 69% | YES |
| Xception | 68% | 76% | 63% | YES |
| Inception v2 | 82% | 82% | 75% | No |
| Inception v3 | 81% | 84% | 72% | YES |
| ResNet50 | 85% | 89% | 83% | No |
| ResNet101 | 82% | 85% | 79% | No |
| VGG16 | 88% | 91% | 84% | No |
| VGG19 | 88% | 90% | 85% | No |

Finally, for the combined datasets, we use all information gathered from our three datasets to create an overall picture of the accuracy of our CNN models, as well as how each handles potential gender bias. Comparing Tables 6 and 7, the accuracy of the CNN models with this combined dataset is generally lower than with the SiW Database. This could be due to a plateauing of accuracy levels from increasing the overall number of images that are trained and tested. However, the accuracy of the combined dataset is higher than both the WMCA Dataset and the CASIA-FASD Dataset.

As can be seen in the last column in Table 8, the disparity between the accuracy results for the male dominant and female dominant image subsets is greater than 10% for 6 out of 8 of our models, meaning that it is very likely that those models are affected by gender bias. While the gender bias levels for both VGG16 and Resnet50 does not reach the 10 percent disparity threshold, differences of 5 percent and 8 percent, respectively, show that both models are not immune to potential biases. Due to these combined results, VGG16 and ResNet50 look to be very strong candidates for the creation of a hybrid CNN model that has both a high accuracy level, as well as resistance to gender bias.

*4.2. Conclusions*

There were several clear winners in our experiments, with respect to both accuracy as well as gender bias resistance. VGG-16 in particular showed great accuracy across all datasets and image subsets scoring the highest out of all of our tested CNN models in the majority of our testing. In addition, the VGG-16 architecture continued to have a gender disparity of less than 10 percent across all tests that were conducted. On the other hand,

while VGG-19 attained accuracy nearly on par with its counterpart, that model stumbled with the WMCA and combined datasets where gender bias resistance is concerned.

**Table 8.** Accuracy of tested CNNs by gender, combined datasets, 80/20 split.

| CNN Model | Balanced | Male Dominant | Female Dominant | Disparity > 10%? |
|-----------|----------|---------------|-----------------|------------------|
| AlexNet | 75% | 78% | 65% | YES |
| Xception | 69% | 74% | 59% | YES |
| Inception v2 | 81% | 83% | 72% | YES |
| Inception v3 | 79% | 83% | 70% | YES |
| ResNet50 | 84% | 88% | 80% | No |
| ResNet101 | 82% | 83% | 71% | YES |
| VGG16 | 89% | 91% | 86% | No |
| VGG19 | 86% | 87% | 76% | YES |

ResNet50 was another bright spot for our CNN models. While having accuracy that was slightly lower than either VGG model, ResNet50 was able to attain good ratings with the gender disparity testing, with results no worse than 8 percent. Due to these results, in future work concerning this topic, VGG-16 and ResNet50 should be put forth in the attempt to create a hybrid CNN model for the purpose of mitigating gender bias.

Unfortunately, AlexNet and Xception performed poorly in all of our testing when compared to the other CNN models. Not only did they show lackluster accuracy levels, they also showed the potential for gender bias in every single test conducted. While the Inception models showed better accuracy than AlexNet or Xception, they too showed that they were not immune to potential gender bias, having a gender disparity score of greater than 10 percent in all but one test that were conducted.

### 4.3. Limitations and Future Work

There is great opportunity for future work with the results from these CNN models. Through our testing, we have indeed confirmed that gender bias in facial recognition systems is a reasonable concern that needs to be adequately addressed. Working with the most accurate and bias resistant models in our study, VGG-16 and ResNet50, we propose a novel hybrid model combining those two architectures, which will potentially mitigate gender bias, whilst continuing to show high accuracy for facial image recognition.

Additionally, for any future testing of this hybrid model (or other CNN models) to take place, it is of paramount importance that we investigate and implement new video or image datasets that contain various demographic labels. It is only with these enhanced datasets that we will be able to accurately test various CNN models to determine if they are able to resist or withstand potential gender or demographic bias in the future.

There is an opportunity available to improve the results of our work with various new and emerging technologies, including Multispectral Facial Recognition [44]. This technology makes use of both visible and infrared images to enhance the abilities of various Deep Neural Networks [45]. By using this and other new state-of-the-art technologies, we will continue to improve our results and reduce the disparity between accuracy levels of different genders.

Speaking of improvements, this work could benefit greatly from 5-fold cross validation, versus the current methods utilized for examining the datasets. 5-fold cross validation could see improvements in both the accuracy levels, as well as the overall robustness of the methodology. In future work with this topic, as well as with future datasets and CNN methods, we will be utilizing 5-fold cross validation to obtain our experimental results.

Other limitations in our work include a lack of datasets with specific demographics labels, which we have previously discussed, along with accuracy levels that are below what is typical for our tested CNNs. This is due to the specific way that we conducted our CNN testing, choosing to focus more on how our CNNs were able to handle different genders, instead of simply on their overall accuracy as a whole. In future work with the

topic, we plan to combine both state-of-the-art techniques and emerging demographic bias mitigation methods to both improve the overall accuracy of our CNNs, as well as produce more equitable results for all tested individuals.

In addition to the previously mentioned limitations, there were several delays that affected the amount of time needed to fully conduct a range of experiments on the data. As such, some planned experiments on the datasets and CNN models were removed. It is the author's goal to conduct further experiments with these and other CNN models, using new and existing datasets in the future.

Finally, even as this research is funded and published, new CNN architectures are being created and implemented, some which no doubt have the propensity for high accuracy with gender bias resistance. It is our duty to seek out these new CNN models and utilize them for testing and comparison against our newly proposed VGG-16/ResNet50 hybrid CNN model.

**Author Contributions:** Conceptualization, T.G., K.R.; methodology, T.G.; software, T.G.; validation, T.G.; formal analysis, T.G.; investigation, T.G. and K.R.; resources, T.G.; data curation, T.G. and K.R.; writing—original draft preparation, T.G.; writing—review and editing, K.R.; visualization, T.G.; supervision, K.R.; project administration, K.R.; funding acquisition, K.R. All authors have read and agreed to the published version of the manuscript.

**Funding:** This research is supported by National Science Foundation (NSF). Any opinions, findings, and conclusions or recommendations expressed in this material are those of the author(s) and do not necessarily reflect the views of NSF. Award Abstract #1900187 https://nsf.gov/awardsearch/showAward?AWD_ID=1900187, accessed on 1 September 2022.

**Institutional Review Board Statement:** Ethical review and approval were waived for this study, due to the fact that publicly available datasets were used. WMCA Dataset—idiap.ch/en/dataset/wmca; CASIA-FASD Dataset—paperswithcode.com/dataset/cassia-fasd; SiW Database—cvlab.cse.msu.edu/siw-spoof-in-the-wild-database.html, accessed on 1 September 2022.

**Informed Consent Statement:** Patient consent was waived due to the fact that publicly available datasets were used. WMCA Dataset—idiap.ch/en/dataset/wmca; CASIA-FASD Dataset—paperswithcode.com/dataset/cassia-fasd; SiW Database—cvlab.cse.msu.edu/siw-spoof-in-the-wild-database.html, accessed on 1 September 2022.

**Data Availability Statement:** All data for this study is currently unavailable pending research maturity.

**Conflicts of Interest:** The authors declare no conflict of interest.

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
