# Peer review of "Examining Gender Bias of Convolutional Neural Networks via Facial Recognition"

_futureinternet, doi:10.3390/fi14120375_

Round 1
Reviewer 1 Report
In this paper, the demographic bias in facial expression recognition applications is addressed. Several CNNs are used and then a new CNN method is proposed, comprised of the best parts of several tested CNNs, which aims to be both highly accurate, as well as able to mitigate demographic bias across a substantial range of individuals and groups. Before accepting the paper, the following major comments must be addressed carefully and minor comments are added directly to the manuscript.
1) There are long sentences, which can be improved.
2) For the abstract part, the author should claim the background, or pose the problem/motivations and the contributions.
3) Add a section Motivations and contributions where you describe the motivations behind this work and then your contributions.
4) The literature should be up to date. Please include the following articles in the reference: Schoneveld et al. (2021) : Leveraging recent advances in deep learning for audio-visual emotion recognition, SKW Hwooi et al (2022) about Deep Learning-Based Approach for Continuous Affect Prediction From Facial Expression Images in Valence-Arousal SpaceDOI: 10.1109/ICIP42928.2021.9506025 and https://doi.org/10.1007/s11042-020-10332-7
5) More experiments are required to validate the proposed approach
6) the performance of your approach must be compared with existing methods on the three used datasets in your experiments.
7) The author should provide the code in the revised version.
8) Experiments: Hold-out method is always biased in terms of performance comparison. It requires k-fold cross validation. Add statistical analysis.
9) Add a new section where you discuss the limitations of your work
10) Figures should be created from scratch. Colors should be standardized (in the whole paper), use one font for all things. Moreover, the quality of the images must be higher.
11) the paper does not read well. The authors are advised to carefully review the paper to improve its readability. The paper should be revised by an English native speaker. There are a lot of English mistakes.

Reviewer 2 Report
The main objective of journal Future Internet (ISSN 1999-5903) is science and research concerned with evolution of Internet technologies. Consequently, the submitted paper, being a facial recognition work, appears to be off-topic.
The paper under review does not include Multispectral Facial Recognition (see https://dx.doi.org/10.1109/ACCESS.2020.3037451 and https://doi.org/10.3390/s21134520 ).
A state of the art with only 3 paragraphs is too small. On the other hand, the articles in the state of the art are from the year 2020 and older, which suggests an aging state of the art.
It is not appropriate to be publishing figure 5 when this figure has already been published in the article cited in [40]. All figures are expected to be new. The same happens in figure 7 with reference [24], figure 8 and reference [25].
The authors tested several CNN algorithms, on various datasets, describing in some detail each algorithm/network used. As the details of each CNN have already been published in the literature, it is more important for the authors to point out the differences they made when compared to the original implementation.
Correct the legend in Figure 6. Infrared is a very generic word, as it can be near infrared, shortwave infrared, mediumwave infrared or longwave infrared also known as thermal image.
With the objective of testing CNN models to discriminate demographic bias, the authors decided to analyze gender bias.
The authors do not write any scientific explanation for the performance of CNN algorithms being less than 100%, reaching values close to 50% in some cases, which is bad. It is suggested that the authors pay special attention to the face detection module, increasing the number of landmarks on each face, in order to improve the overall performance of face recognition.
Round 2
Reviewer 2 Report
In the first version of the paper, this reviewer wrote several comments and questions.
The authors answered all questions and made changes to the paper.
The final quality of the paper has improved.
Author Response
Thank you for your feedback. Your assistance in reviewing this manuscript has helped to improve the quality of this paper.
Respectfully,
Tony Gwyn